# Low-Carbon Travel Motivation and Constraint: Scales Development and Validation

**DOI:** 10.3390/ijerph19095123

**Published:** 2022-04-22

**Authors:** You-Yu Dai, An-Jin Shie, Jin-Hua Chu, Yen-Chun Jim Wu

**Affiliations:** 1International Business School, Shandong Jiaotong University, Weihai 264209, China; psyyyt@gmail.com (Y.-Y.D.); lncccjh@163.com (J.-H.C.); 2College of Business Administration, Huaqiao University, Quanzhou 362021, China; 3School of Big Data, Fuzhou University of International Studies and Trade, Fuzhou 350200, China; 4International College, Krirk University, Bangkok 10220, Thailand; 5Graduate Institute of Global Business and Strategy, National Taiwan Normal University, Taipei 106, Taiwan; wuyenchun@gmail.com; 6College of Humanities and Arts, National Taipei University of Education, Taipei 106, Taiwan

**Keywords:** low-carbon travel motivation, low-carbon travel constraint, low-carbon travel behavior, independent tourist, scale development

## Abstract

Low-carbon travel has emerged as a topic of interest in tourism and academia. Studies have offered reasons tourists may engage in low-carbon travel; however, these explanations are scattered throughout the literature and have yet to be integrated into low-carbon travel motivation and constraint constructs. This study develops a low-carbon travel motivation scale (LCTMS) and a low-carbon travel constraint scale (LCTCS). It performs reliability and validity testing to measure the low-carbon travel motives and obstacles. Items were collected primarily by literature review, and, then, by surveys of 382 tourists from low-carbon travel destinations and 390 from non-low-carbon travel destinations. Through a rigorous scale development process, this study identifies six dimensions of the LCTMS (environmental protection, experience-seeking, escape or social connection, industry pleas and measures for environmental protection, low-carbon products, and green transportation) and four dimensions of the LCTCS (intrapersonal constraints, interpersonal constraints, structural constraints, and the not a travel option).

## 1. Introduction

Carbon dioxide emissions resulting from travel and tourism activities account for 5–14% of the world’s total carbon dioxide emissions (World Tourism Organization and United Nations Environment Programme, 2012). They are rising at a rate of 3.2% per year [1]. Transportation (e.g., aircraft, cars, and boats), accommodation, and tourism activities represent the primary means tourism consumes energy and produces carbon emissions [2,3]. Jarratt and Davies [2] have asserted that tourists could slow the rise in carbon emissions, by consuming less fuel and reducing their carbon emissions. Thus, promoting low-carbon travel has become a topic of interest to the tourism industry and academic circles over the past ten to twenty years.

The push and pull factors commonly studied in tourism motivation research [4] serve as a sound theoretical basis for determining why tourists engage in low-carbon travel. Push factors are the socio-psychological forces that drive people to engage in low-carbon travel, such as reducing the harmful effects of tourism on the environment [5], recognizing the concept of energy-saving and carbon-reducing (ESCR) [6], and enjoying healthy movement [7]. Pull factors are the aspects of low-carbon travel activities that attract tourists, such as reducing one’s carbon footprint [5], providing eco-friendly products and food [8], offering ESCR tourism facilities and consumer discounts [9], and using operators that hold environmental certifications [6]. Although studies have postulated various reasons for visitors potentially participating in low-carbon travel, the explanations are scattered throughout the literature. They have not been integrated into a unified theory of low-carbon travel motivation.

On the other hand, most tourists recognize the benefits of low-carbon activities but are reluctant to plan low-carbon travel activities [10]. The key to influencing tourists’ decisions is travel constraints that act as barriers and prevent them from traveling in general or traveling to the extent they would like [11]. Travel constraints can be defined as those that inhibit continued traveling, cause the inability to travel, result in the failure to maintain or increase the frequency of travel, and/or lead to negative impacts on the quality of the travel experience [12,13]. They prevent decision-makers from engaging in travel, even though the motivation may exist.

Very few studies have integrated and developed scales for two constructs in a sole paper [14,15]. However, Wen et al. [14] suggested that creating valid and reliable multiple-item self-report scales is a significant priority for marketing and consumer research. Božić et al. [15] mentioned: “Understanding what motivates and hinders people from traveling has important practical implications, as it helps better understand and predict travel decisions and consumption behavior of tourists”.

Therefore, this study identifies and clarifies low-carbon travel motivations and low-carbon travel constraints, from a theoretical perspective of the two concepts, and adds to the related research. These low-carbon travel motivation and low-carbon travel constraint constructs can help researchers and tourism operators understand the reasons and limitations of tourists engaged in low-carbon travel, while also providing appropriate market segmentation criteria in tourism. Understanding the motives and constraints of low-carbon travel will help the tourism industry provide the services that tourists want and need, and assist the operators in planning their corresponding marketing strategies.

Based on the background above research and topics, the objectives of this study are as follows: (1) To explore the motives and obstacles to engaging in low-carbon travel, review the literature to identify their significant implications, and then clearly define constructs for low-carbon travel motivation and low-carbon travel constraint. (2) To develop a low-carbon travel motivation scale (LCTMS) and a low-carbon travel constraint scale (LCTCS) and, then, perform letter and validity testing to measure the low-carbon travel motives and obstacles.

## 2. Literature Review

### 2.1. Low-Carbon Travel Motivation

Travel motivation is a broad and complex concept. McIntosh and Goeldner [16] divided primary travel motivations into four types: physical motivation, cultural motivation, interpersonal motivation, and prestige motivation. Pearce et al. [17] listed ten primary tourist motives. Experience the environment, rest and relax in a comfortable site, pursue special interests or skills, be healthier, and possess a strengthened physique, are push or internal motivation. Interact with locals, understand the local culture, improve family life, gain self-protection, gain security, be respected, win social status, and reward selves are pulled or external motivation.

As visitors in different travel situations may hold different motivations, researchers have used many aspects to discuss tourism motivation and develop different travel motivation scales. For example, rural tourists are generally motivated by relaxation, social interaction, education, family gatherings, novelty, and excitement [18]. Cruise tourists are usually motivated by self-esteem and social identity, exploration and relaxation, learning and discovery, novelty and stimulation, and socialization and cohesion [14]. Slow travelers are generally motivated by relaxation, introspection, escape, novel pursuit, participation, and discovery [19].

Although these researchers have mainly used factor analysis or cluster analysis to determine the various travel motivations of their subjects, since the nature of travel is a series of travel activities to satisfy people’s inner social-psychology needs or the external cultural seeking of a destination, tourist motivation has tended to revolve around the concepts of “pull” and “push” [20]. Most discussions in tourism have applied the theory of push-and-pull motivation when explaining why people travel [21,22,23,24]. Therefore, the push and pull factors of Crompton [20] provide the main theoretical framework in this study.

Researchers have proposed numerous factors that motivate tourists to engage in low-carbon travel. However, these explanations are scattered throughout the literature, have not been integrated into a low-carbon travel motivation construct, and are inconsistent with general tourism motivations. To address these discrepancies, this study applies the push and pull theory [4] to explore the factors that motivate tourists to participate in low-carbon travel.

#### 2.1.1. Push Factors

Push motivation is a psychosocial need that motivates individuals to travel, and that motivation drives or guides individual travel choices [25]. Common push factors include knowledge, relaxation, family harmony, escape, self-discovery, prestige, and social interaction. According to the target framework theory of Lindenberg and Steg [26], we can explore motivation related to environmental protection behavior from three aspects: morality, emotion, and access.

Moral motivation refers to tourists participating in environmental protection behavior based on their social awareness of green products and service consumption as well as a positive self-image. Tourists choose low-carbon destinations because of environmental appeal or green policies promoted by the tourism industry [6]. When Horng et al. [6] applied self-completion theory, they discovered that consumers’ inner moral identity and symbolic moral identity drove them to choose low-carbon options. Several of the subjects interviewed by Dickinson et al. [27] said they decided to travel by train out of concern for the environment. Horng et al. [8] discovered that tourists make eco-friendly decisions due to their sense of responsibility for the environment. Thus, morality appears to be a factor that motivates tourists to engage in low-carbon travel.

Emotional motivation mainly refers to the joy [28] and happiness that tourists receive from altruistic behavior [29]. Nawijn and Peeters [30] asserted that green consumers are motivated by altruism and the resulting joy. Due to expected benefits for the environment and the next generation, visitors consume low-carbon products and services from green restaurants and low-carbon players [31]. Kuo and Dai [5] have determined that, once they have recognized that travel hurts the environment, tourists begin to select low-carbon travel options to sustain the environment and recreational resources of the destination, lessening the environmental problems caused by individuals and society. Applying protection motivation theory, Horng et al. [8] determined the ESCR behavior of tourists motivated by threat assessments, such as environmental risk severity and vulnerability, and adaptation assessments, such as response effectiveness and self-efficacy. The study showed that tourists engage in ESCR tourism because they believe ESCR behavior helps protect the environment. Thus, the desire to preserve the environment of a destination and preserve its recreational resources for the next generation appears to be one reason tourists choose low-carbon travel.

Access motivation refers to tourists’ desire to improve physical fitness and acquire new knowledge. Regarding physical fitness, Garrido-Cumbrera et al. [7] used focus groups to determine that tourists who choose to walk or cycle are driven by a strong motivation to exercise. Travelers interviewed by Kuo and Dai [5] said that to promote their health, they would shift from general tourism activities to low-carbon travel activities such as cycling, walking, and eating more vegetables and less meat. As for learning, tourists are motivated to acquire new ESCR knowledge. Horng et al. [8] modified the tourist learning motivation scales of Ballantyne et al. [32] and proposed that tourists are motivated to participate in ESCR festivals. Motives include: acquiring new ESCR practices, obtaining more ESCR information, expanding more interesting ESCR topics, inspiring the willingness to implement ESCR, and discovering new ESCR methods. Therefore, personal fitness and the desire to acquire new knowledge motivate tourists to participate in low-carbon travel.

#### 2.1.2. Pull Factors

Push motivations are usually intrinsic, whereas pull motivations are external and are related to destination choice, destination traits, attractions, and attributes. Pull factors attract people to destinations such as sunny beaches or snowy mountains and influence the perceptions and expectations of travelers. Common pull factors are natural and historical environments, cost, convenience, safety, accessibility, novelty, and education [20].

ESCR characterizes low-carbon travel. Generally, tourists engage in low-carbon travel activities to directly reduce carbon emissions and their carbon footprint while on holiday [5,10]. Moreover, visitors are attracted by low-carbon travel-related food, transportation, and accommodations. Researchers in the hotel [6] and restaurant industries [31] have confirmed that visitors generally favor companies with environmental certifications. Eco-friendly policies and actions, such as not changing sheets and towels daily, garbage sorting, and resource recycling, can also attract tourists. Tourists may also seek alternative methods to protect the environment. Thus, tourists are willing to accept ESCR technologies to engage in ESCR activities.

Some tourists reduce their travel by plane once they understand the importance of reducing their carbon footprint [27]. Garrido-Cumbrera et al. [7] discovered that tourists often walk or cycle to enjoy the outdoors. Exposure to eco-friendly products, facilities, and food also affects travelers’ ESCR behavior during a trip. For example, hotels might offer ESCR rooms and facilities [6]. Low-carbon tourists seek simple packaged products and locally produced fresh foods [33]. Finally, discounts to incentivize eco-friendly consumption also motivate travelers to change their behaviors [9,33]. Each of these factors can help achieve ESCR.

Research by Kuo and Dai [5] revealed that past low-carbon travel experiences substantially influence future decisions to participate in low-carbon travel. For example, some travelers choose to engage in low-carbon activities such as walking or cycling because past experiences were relaxing and enjoyable [7]. Horng et al. [6] showed that consumers’ past dining experiences could stimulate green consumption behavior in line with ESCR. Chen [9] pointed out that travelers who have previously stayed in green hotels make green accommodation decisions based on their past accommodation experiences. Thus, the characteristics that attract tourists to low-carbon travel include the ESCR characteristics and measures of the trip, residence, food, shopping, and tourists’ past travel experiences.

#### 2.1.3. Summary

Empirical research has focused on identifying tourism motivation factors and measurement scales. A variety of different motivational factors have been used to explain the behavior of visitors in different travel situations. In the past, researchers proposed a variety of motivations to explain why tourists engage in low-carbon travel. However, no research has integrated this into the low-carbon travel motivation construct. Understanding tourists’ low-carbon travel motivations will help us identify the factors that drive tourists to participate in low-carbon travel. Industry operators will be better able to plan low-carbon travel activities to maximize tourist satisfaction, inspire and change attitudes, and reduce environmental stress [8]. Thus, both tourists and operators can more effectively develop low-carbon travel to achieve sustainability.

### 2.2. Low-Carbon Travel Constraint

Travel constraint is a multifaceted concept that refers to the factors that prevent or reduce the frequency, proportion, or fun of individuals engaging in specific activities [15,34]. Khan et al. [13] argued that travel constraint is a crucial factor that prevents people from starting or continuing to travel. Wen et al. [14] defined travel constraints as factors that inhibit travel continuity, lead to the inability to travel, fail to maintain or increase travel frequency, and negatively influence travel quality. Karl et al. [12] expanded the definition to obstacles that hinder the continuing use of leisure services, the inability to participate in new activities, the inability to maintain or increase the frequency of participation, and the negative effect on leisure experience quality.

There are a few classic conceptual models of barriers to leisure participation. The leisure constraints model developed by Iso-Ahola and Mannell [35] placed the individual in the social environment, but did not explicitly indicate the process by which barriers may operate beyond the individual. This paradigm of leisure constraints also failed to anchor these constraints within the context of the preference–participation relationship [34]. To understand the barriers or reasons that prevent people from traveling, we should, comprehensively, consider the tourists’ perceived constraints or switching barriers. The most common and comprehensive framework in leisure research is the three-dimensional constraint construct proposed by Crawford and Godbey [36] and Crawford et al. [37]. This model assumes that constraints could be dynamic and intervening factors that affect an individual’s participation and preference [38].

Tan [39] explained that research on leisure barriers and travel constraints has often been based on a model of intrapersonal constraint, interpersonal constraint, and structural constraint. Crawford et al. [37] asserted that an individual’s intrapersonal constraints are related to their mental state, including personality traits, attitudes, beliefs, and emotions. Interpersonal constraints are determined by interactions with friends, family, colleagues, neighbors, and others. Structural constraints prevent people from acting, including economic resources, available time, and accessibility. Therefore, the low-carbon travel constraint base is also in line with Crawford et al.’s [37] model. Given the complex nature of travel constraints, researchers often use the three-dimensional leisure constraint framework to explore the obstacles of tourists in different travel situations. These include essential natural tourism [40] and cruise tourism [14]. Adopting factor analysis, many travel scholars have confirmed this three-dimensional leisure constraint framework to be effective and reliable [41].

In the low-carbon travel context, researchers e.g., [5,42] have investigated the constraints of low-carbon travel. However, the factors that hinder tourists from participating in low-carbon travel are scattered throughout the literature and have yet to be integrated into a low-carbon travel constraint construct. According to McKercher et al. [10], less than 4% of Hong Kong residents surveyed have adopted low-carbon travel behavior. Juvan and Dolnicar [43] pointed out that even the best intentions of tourists might not be enough to leverage care for the environment into eco-friendly actions. Even though low-carbon travel reduces carbon emissions from travel activities and lessens the adverse effects on the environment, tourists are not always willing to engage in low-carbon travel. To examine the specific reasons that people do not participate in low-carbon travel, this study adopts the three-dimensional travel constraint framework proposed by Crawford and Godbey [36] and Crawford et al. [37]. In addition, we add the concept of the “not a travel option” to explore low-carbon travel constraints.

#### 2.2.1. Intrapersonal Constraints

Poor health and low-carbon travel awareness are the most common intrapersonal constraints. The subjects of Kuo and Dai [5] believed that poor personal health, lack of an environmental sustainability concept, and lack of low-carbon travel information prevent them from engaging in low-carbon travel. Dickinson et al. [44] interviewed slow travelers on low-carbon travel, and most respondents had an insufficient understanding of climate change and, even, doubted the scientific basis. According to Horng et al. [8], international tourists to Taiwan generally do not believe that tourism and related activities are seriously harmful to the environment. Besides, travelers may doubt that individual low-carbon travel can influence climate change [45]. Subjects of Dällenbach [42] argued that an individual’s actions to reduce carbon emissions are inconsequential in a global context.

Along with poor health and lack of awareness, another cause of intrapersonal low-carbon travel constraints is tourists’ unwillingness to sacrifice general tourism’s advantages and personal travel benefits. Respondents of Dällenbach [42] believed that holidays are essential and are benefits that should not be restricted. Dickinson et al. [27] pointed out that tourists would continue to use air travel. Most travelers find it difficult to escape from a high-carbon lifestyle while traveling [45], choosing to act responsibly at home or in other areas [44].

#### 2.2.2. Interpersonal Constraints

Tourism research has determined a lack of friends as the primary interpersonal constraint [40,41]. However, only a few studies have mentioned the interpersonal constraint of lack of friends in the low-carbon travel context. For example, Dickinson et al. [27] pointed out that airline tourists are influenced by family and friends and do not choose low-carbon transportation. The primary interpersonal constraint in low-carbon travel is that tourists believe responsibility lies with others.

According to Dällenbach [42], most tourists believe that governments, businesses, and other countries are the main contributors to climate change. Governments’ continued expansion of airports has raised doubts regarding the government’s determination that people will travel less. The tourism industry has not responded to climate change within its boundaries but has, instead, passed it on to consumers. Therefore, Becken [46] pointed out that tourists believe that reducing carbon emissions is a public, rather than a personal, responsibility. This view has led tourists to be more inclined to spend energy on travel. Therefore, besides a lack of friends, tourists’ interpersonal low-carbon travel constraints also include the belief that other parties bear a greater ESCR responsibility.

#### 2.2.3. Structural Constraints

The characteristics of low-carbon travel, including short-distance travel, less travel, and more extended stays at destinations, are different from general travel planning and are the factors tourists consider most [2,27]. Since low-carbon travel seeks to reduce the high-carbon behavior of available tourism (such as air travel), many tourists believe that engaging in low-carbon travel is difficult, uncomfortable, and inconvenient [3,5].

The structural constraints of low-carbon travel are that its planning and arrangement are different from general tourism. Although all survey participants are aware of climate change, they lack explicit action means. Moreover, many people firmly believe that they are not responsible for carbon emissions related to tourism travel. To a large extent, travelers are limited by tourism products. For example, if it is challenging to take a bicycle on a train, but it is easy to take a bike on low-cost airlines, low-carbon bicycle travelers may engage in high-carbon aircraft travel [44].

#### 2.2.4. Not a Travel Option

Many studies have interviewed travelers who generally do not consider alternative transportation on reducing airplane tours [27,42,44]. Dällenbach [42] pointed out that tourists believe that the price and time of traveling on trains are not as valuable as travel by air. Traveling by plane is the preferred and customary tourist choice, and low-cost airlines enable people to travel more overseas. Dickinson et al. [27] pointed out that aviation tourists generally have no other transportation options, and, therefore, low-carbon travel may not be a tourist’s first option.

#### 2.2.5. Summary

Travel constraints have been studied extensively in the field of tourism. Although many tourists refuse to embrace low-carbon travel, studies have not integrated the various constraints into a construct of low-carbon travel constraint. Therefore, this study applies the three-dimensional constraint architecture, proposed by Crawford and Godbey [36], to clarify the low-carbon travel constraint. Researchers believe that tourists’ low-carbon travel constraints explain why tourists with strong low-carbon travel awareness do not engage in low-carbon travel. Helping them overcome those constraints can encourage them to adopt low-carbon travel. In addition, given that travel constraints have been used as variables in the tourism market, the development of LCTCS allows researchers to precisely measure the obstacles preventing tourists from engaging in low-carbon travel and segmenting the low-carbon travel market [47].

## 3. Methodology

This study aims to explore the motivations and constraints of tourists engaged in low-carbon travel and to develop LCTMS and LCTCS. This study mainly uses a literature review to collect tourists’ potential low-carbon travel motivations (based on push-and-pull factors) and low-carbon travel constraints (intrapersonal, interpersonal, and structural). It adopts interviews to supplement more possible items. The researcher and the transcribed responses conducted in these interviews would be coded, and a grounded theoretical context would be used to guide the coding process [48]. Moreover, this study follows the broad-scale development process and constructs a reliable and valid measurement tool. This study builds two scales following the scale development processes of Churchill [49], Ager [48], and Bhatt et al. [50]. Please see Figure 1.

### 3.1. Generating Initial Items

#### 3.1.1. Principles and Form

Schmid [51] suggested using a multifaceted approach to produce a more comprehensive list of measurement items. Therefore, this study uses three techniques. First, a series of measurement items are generated from a complete literature review. This study first reviews the literature and, then, lists tourists’ low-carbon travel motivations and constraints. The literature review focuses on research papers and books on low-carbon travel. Churchill [49] and Bhatt et al. [50] stated that researchers should strictly specify which definitions they retain and exclude. Therefore, the literature reviewed must list why tourists engage in low-carbon travel and the key factors that prevent tourists from starting or continuing low-carbon travel. The articles reviewed must express or imply conceptual descriptions of definitions of low-carbon travel motivations and constraints. For the empirical research literature, the researcher’s choice of reference material must be based on the nature of the research sample, the field in which the data were collected, and the activities undertaken by tourists. Therefore, this study reviews data types consistent with the low-carbon travel motivation and constraint concepts. This study adopted the translation and back-translation process [52]. We reviewed original items from the English literature, translated them into Chinese, and ensured the responders understood the meanings of each item, such as ESCR.

#### 3.1.2. In-Depth Interviews and Content Analysis

To collect a variety of low-carbon travel motivations and low-carbon travel constraints as well as to improve data quality and generalization, this study adds some items through interviews. From 1 February to 31 March 2015, 14 low-carbon tourists and 11 non-low-carbon tourists participated in in-depth interviews by snowball sampling. The open-ended interview questions are shown as Appendix A. The respondents’ characteristics satisfy the following conditions: (1) males and females are similar; (2) ages are 21 to 30 years old, 31 to 40 years old, and more than 40 years old; (3) travel experience covers less than 5 years, 6 to 10 years and more than 10 years; (4) the low-carbon travel experience includes those that have participated in group trips of low-carbon tourism, have planned low-carbon travel journey, and have practiced ESCR in tourism.

Next, three judges who have rich low-carbon travel experience and two judges who have research experience in low-carbon travel are invited to assist in classifying and naming the results of the literature review and in-depth interviews. The content analysis produced 113 valid low-carbon travel motivation units and 115 valid low-carbon travel constraint units. These units were grouped into 33 low-carbon travel motivation-related and 30 low-carbon travel constraint-related subcategories. The inter-judge reliability is 0.87 (98/113) and 0.91 (30/33) for low-carbon travel motivation and 0.88 (101/115) and 0.90 (27/30) for low-carbon travel constraint. These low-carbon travel motivation and constraint subcategories were then transformed into preliminary items for the initial item pool.

The researchers also carefully review the conceptual narrative of the above items, so that the item content conforms to the appropriate words of low-carbon travel motivation and low-carbon travel constraint, to achieve reasonable surface validity. The results indicate that low-carbon tourists and non-low-carbon tourists have different motivations and constraint experiences. For clarity, 33 low-carbon travel motivations are divided into two categories: push motivations and pull motivations. The 30 low-carbon travel constraints are divided into four categories: intrapersonal constraints, interpersonal constraints, structural constraints, and the not a travel option.

#### 3.1.3. Expert Validation

After pooling the initial items, the researcher submits the preliminary research to an expert panel of four experienced low-carbon tourists and five scholars specializing in tourism and low-carbon travel for assessment. Chang et al. [53] stated that items not sufficiently consistent with their construct should be modified or excluded from the scale. Thus, the research is submitted to nine low-carbon travel experts to assess the applicability of these measurement items and assist in screening the clarity and surface validity of the questionnaire questions, which helped make the items more accurate [54]. This study then adds, merges, and deletes questions, and modifies the semantics to form an initial questionnaire. Finally, 26 low-carbon travel motivations and 23 low-carbon travel constraints are retained.

The formal questionnaire is based on previous methods of measuring travel motivations and constraints. This study uses the intensity concept (e.g., Wen et al. [14]) to measure tourists’ low-carbon motivations and constraints. The questionnaire uses a 5-point Likert Scale from 1 (strongly disagree) to 5 (strongly agree). The formal questionnaire also gathers respondents’ gender, age, education level, marital status, occupation, monthly income, place of residence, and past low-carbon travel experience.

#### 3.1.4. Item Analysis

To increase the reliability and validity of the measurement tools, this study uses SPSS 23.0 software to individually evaluate each item, confirm that the items measure the same concept and form an intrinsically consistent scale, and exclude items that do not meet this standard [55]. The corrected item–total correlation is used to analyze the items in both scales of this study [56]. The analysis reveals high discrimination and homogeneity in low-carbon travel motivation and low-carbon travel constraint items.

### 3.2. Pre-Test and Item Refinement

To ensure the ability of the questionnaire to measure low-carbon travel motivations and low-carbon travel constraints, this study conducts a pre-test to confirm the clarity and comprehensiveness of the questionnaire content. The basis of filtering items is item analysis and a critical test. Item analysis can eliminate ambiguous or misleading items in the scale development process. In this way, researchers can consider whether items on a scale are necessary or not [57]. This study, preliminarily, surveys college students in universities’ tourism and leisure departments in Weihai City, Shandong Province, China. In addition to being convenient, this sampling method has two benefits. First, these students may or may not engage in low-carbon travel. Second, as Schwall et al. [58] argued, student samples are relatively homogeneous, so the results are less likely to obscure the scales testing. The pre-test data collection occurred between 26 May and 31 May 2017. Based on the pre-test results, two motivation items are deleted since their t-test results in the critical test are not significant, including “I am engaged in low-carbon travel because of I can save on travel expenses” and “I am engaged in low-carbon travel because of the prices of the products are reasonable”. Subsequently, a formal questionnaire features 24 low-carbon travel motivations and 23 low-carbon travel constraints.

## 4. Results

### 4.1. First Formal Survey

#### 4.1.1. Data Collection

The first formal data collection covers low-carbon destination visitors. Low-carbon destination refers to destinations with low-carbon footprints and low-carbon travel concepts, such as travel planning and merchant services [59]. Qingdao is an important coastal tourist city in eastern Shandong, China, with rich and diverse tourism resources and an excellent ecological environment. During the rapid development of urbanization, industrialization, and tourism, Qingdao has formulated several environmental protection policies such as “Opinions on Implementing the Scientific Outlook on Development and Strengthening Environmental Protection”, and carried out many environmental protection projects such as the “Clear Water and Blue Sky Project” to promote the construction of ecological civilization in many aspects, such as industrial structure upgrading, energy-saving, and emission reduction, as well as promotion of low-carbon tourism concepts. In recent years, Qingdao City has successively won the title of China’s Travelable, Livable, and Industrial City. Local businesses must comply with various rules: (1) the lodging industry must not provide single-use toiletries; (2) the catering industry must not provide disposable tableware and, instead, must encourage visitors to bring their tableware; and (3) travel agencies must remind and encourage visitors to comply with the low-carbon island lifestyle.

Independent tourists generally plan and arrange their travel, holding a wealth of knowledge regarding the purchase and use of travel products. Independent tourists desire to satisfy their inner social and psychological needs through tourism. So, compared to group tourists, independent tourists are much more attracted by tourism attributes. Independent tourists also consider the constraints of planning travel routes more than group travelers. Therefore, independent tourists require more time and energy to decide whether to start or continue low-carbon travel. This study selects Laoshan Scenic Area and Zhanqiao Scenic Area in Qingdao as the survey sites for low-carbon destinations. The survey population is independent tourists visiting these scenic areas. This study adopts random sampling to ensure the sample is representative.

Survey interviewers were two doctoral students, three master’s students, and two college students with considerable investigative skills training. The researchers used the convenience sampling method from 1 June to 14 June 2018. To ensure the validity of the sample, interviewers reviewed each item in the collected questionnaires to ensure that respondents answered all questions. This study collected 497 questionnaires, of which 382 were valid, for an effective sample recovery rate of 76.9%. Roscoe [60] suggested that the number of samples is 10 times the number of variables. In this study, the ratio of samples to low-carbon travel motivation items is 15.9:1, while the ratio of samples to low-carbon travel constraint items is 16.6:1.

#### 4.1.2. Descriptive Statistics Analysis

Data from the first survey are used to purify the LCTMS and LCTCS to determine the scale’s factor structure. Data from the second survey are used to revalidate the stability of the scale factor structure and perform a reliability and validity analysis. The sample and travel characteristics of the low-carbon tourist sites collected in the first survey are detailed in Table 1. Most respondents are female (53.1%), and most are aged 21–30 years (43.5%). The highest educational level of most respondents is college or university (66.8%); most respondents are single (65.4%); the most common careers are student, family management, and retiree (60.2%); and the most common monthly income is more than RMB 5000 (79.6%). Most respondents have experienced low-carbon travel once in the past year (69.4%), and the most common travel time is two days (74.1%).

Normality assumptions were examined using skewness-kurtosis tests. Under univariate normality, we consider variables with a skew index higher than 3 and a kurtosis index higher than 10 to violate the usual distribution assumption (Kline, 2005). The study items were assessed by determining the range of skewness (−1.43 to 0.26) and kurtosis (−0.79 to 2.32). Thus, the first formal survey data did not appear to violate the assumptions of normality.

#### 4.1.3. Exploratory Factor Analysis

This study uses exploratory factor analysis (EFA) on low-carbon travel sites to determine the measurement scale constructs. The principal component analysis method obtains the co-interpreted variation between all measurement items and excludes items with cross-load quantity, by the orthogonal rotation axis of the maximum variation number. Items with factor loading values above 0.5 are retained. The EFA results of the LCTMS are presented in Table 2. Based on these results, this study determines six factors: environmental protection, environmental protection appeals and measures, escape and social connection, low-carbon products, green transportation, and experience seeking. These six factors explain 69.56% of the variance. The Kaiser–Meyer–Olkin (KMO) is 0.94, indicating that the low-carbon travel motivation sample data have a sufficient internal correlation between the conducting EFA.

The EFA results for the LCTCS are presented in Table 3. The study affirms four factors: structural constraints, the not a travel option, intrapersonal constraints, and interpersonal constraints. These four factors explain 60.12% of the variance. The KMO value of 0.92 indicates that the sampling data of low-carbon travel constraints have a sufficient internal correlation between the conducting EFA.

#### 4.1.4. Reliability and Validity Analysis 

Reliability is generally reflected in the consistency or stability of the measurement tool, and it refers to the trustworthiness of the measurements. Table 2 and Table 3 demonstrate that Cronbach’s α values for all factors of LCTMS and LCTCS are more significant than 0.6 [61,62,63]. The overall reliability of the LCTMS is 0.94, and that of the LCTCS is 0.92. These results indicate that the internal consistency of the LCTMS and the LCTCS is, statistically, reasonably high.

Criterion-related validity is used to test the validity of LCTMS and LCTCS. Carvache-Franco et al. [23], and Božić et al. [15] have demonstrated that their motivations and constraints affect tourists’ travel intentions. Their low-carbon travel intentions can test tourists’ low-carbon travel motivations and constraints. Therefore, this study refers to the low-carbon travel intention scale Horng et al. [54] proposed. After the experts’ discussion, three survey statements were retained: “I would like to participate in low-carbon travel”, “I will encourage others to engage in low-carbon travel”, and “I will engage in low-carbon travel myself”.

The Pearson Correlation between intention and motivation and intention and constraint is used to evaluate criterion-related validity. The results demonstrate that environmental protection (r = 0.62), experience seeking (r = 0.60), escape and social connection (r = 0.52), environmental protection claims and measures (r = 0.43), low-carbon products (r = 0.47), and green transportation (r = 0.49) are significant positively associated with low-carbon travel intention. At the same time, structural constraints (r = −0.10), intrapersonal constraints (r = −0.19), the not a travel option (r = −0.42), and interpersonal constraints (r = −0.13) are significantly negatively associated with low-carbon travel intention. All these correlation coefficients are significant at *p* < 0.01. Therefore, the criterion-related validity of LCTMS and LCTCS is supported.

#### 4.1.5. Common Method Variance

The present study evaluated the presence of common method variance (CMV) by the Harman single-factor test [64]. The six factors of low-carbon travel motivation and the four factors of low-carbon travel constraint were separately constrained to a single factor, using factor analysis in SPSS. As per the unrotated factor solution, the percentage variance explained by the single factor of low-carbon travel motivation was 44.5%, and the low-carbon travel constraint was 39.8%, lower than 50% [64]. These results confirmed the absence of CMV in the first formal survey of this study.

### 4.2. Second Formal Survey

#### 4.2.1. Data Collection

Through the first data collection and analysis process, this study establishes 24 questions for the LCTMS and 23 questions for the LCTCS. Next, this study evaluates the reliability and validity of the scales in another different sample, as recommended by Churchill (1979). The second phase investigates independent tourists in non-low-carbon destinations. Since all the counties and cities in Shandong are actively developing sightseeing, most tourists engage in mass tourism. This study selects the metropolitan areas of Weihai City in Shandong as representative samples of non-low-carbon destinations. Data were collected from 15 June to 28 June 2018. Four hundred and eighty-five questionnaires were collected, and 390 questionnaires were valid, for an effective sample recovery rate of 82.3%. The ratio of samples to low-carbon travel motivation items is 16.6:1, and the ratio of samples to low-carbon travel constraint is 17.3:1. The items were assessed by determining the range of skewness (−1.17 to 0.32) and kurtosis (−0.59 to 1.02). Thus, the second formal survey data appear in line with the assumptions of normality.

#### 4.2.2. Descriptive Statistics Analysis

The sample profile and travel characteristics of non-low-carbon travel sites are presented in Table 4. Most respondents are female (61.5%) and are aged <20 years (42.8%), followed by those aged 21–30 years (37.7%). The highest educational level of most respondents is college or university (82.3%); most respondents are single (80.5%); the most common careers are student, family management, and retiree (71.0%); and the most common monthly income is more than NTD 40,000 (91.0%). Most respondents reported not experiencing low-carbon travel in the past year (50.3%), followed by one time (23.6%), and the most common travel time is two days (74.1%). Once again, the Harman single-factor test was adopted to analyze CMV here. Furthermore, the percentage variance explained by the single factor of low-carbon travel motivation was 47.6%, and the low-carbon travel constraint was 44.4%, which is below 50% [64]. These results also confirmed the absence of CMV in the second formal survey of this study.

#### 4.2.3. Confirmatory Factor Analysis

Confirmatory factor analysis (CFA) is used to confirm the factors’ structure of the LCTMS and LCTCS, obtained from the previous EFA results. LISREL 8.80 [65] analyzes goodness-of-fit and measurement model estimations. The squared multiple correlation (SMC) value is used to measure the individual reliability of each item. The t-value of each variable is used to determine whether the variable reaches a significant level. The detailed results are presented in Table 5 and Table 6.

In Table 5, the SMC values of the 24 measurement items of the LCTMS are between 0.54 and 0.81, indicating that each measurement item has strong explanatory power. In addition, the t-values of each item are more significant than 1.96, meaning that each item is at a considerable level. The composite reliability (CR) values of each construct of the LCTMS are more potent than 0.6, as prescribed by Pan et al. [65]. This study exhibits a high degree of internal consistency in line with this standard. In Table 6, the SMC values of the 23 measurement items of the LCTCS are between 0.50 and 0.85, while the t-values of each item are more significant than 1.96. The CR values of each construct are under the standard CR > 0.6, also exhibiting a high degree of internal consistency.

Next, this study explores whether the scale’s measurement mode’s absolute and relative fit indicators meet the criteria. In the overall model of the LCTMS, the χ^2^/df is 780.31/237, GFI is 0.86, AGFI is 0.82, NFI is 0.98, NNFI is 0.98, CFI is 0.98, IFI is 0.98, SRMR is 0.05, and RMSEA is 0.08. These evaluation indicators are all acceptable, indicating that the sampling data exhibit high goodness of fit with the structure of the LCTMS, which suggests a proper scale [66]. From Table 6, the CFA of the LCTCS demonstrates that χ^2^/df is 661.38/224, GFI is 0.88, AGFI is 0.86, NFI is 0.93, NNFI is 0.93, CFI is 0.94, IFI is 0.94, SRMR is 0.08, and RMSEA is 0.08. These evaluation indicators are at acceptable levels, indicating that the sampling data exhibit high goodness-of-fit with the structure of the LCTCS, which suggests a sufficient scale.

#### 4.2.4. Convergence Validity and Discriminatory Validity

Convergence validity refers to the observation items used to measure the same construct, and these should have a high correlation with each other. The standard factor loading of all items in the scale is 0.50, and the estimated parameters (t value) are more significant than 1.96 (*p* < 0.05), reaching the statistically significant level. These suggest that the potential variables of this study have ideal convergence validity [67]. Matthes and Ball [63] indicated that the average variation extraction (AVE) for each construct must be greater than 0.5, indicating that the internal consistency of the scale structure is acceptable. The correlation coefficient between pairwise constructs was compared to the square root of the mean variation extract for each factor to test discriminative validity. Table 7 demonstrates that the scale’s measurement models have high discriminatory validity.

#### 4.2.5. Competing Models

This study also examined two competing models for the LCTMS and the LCTCS separately. One is a first-order model, in which there is LCTMS with six main factors (environment protection, experience-seeking, escape/social connection, environmental protection appeals and measures, low-carbon products, and green transportation) and LCTCS with four main factors (structural constraints, intrapersonal constraints, the not a travel option, and interpersonal constraints). The other one is a second-order model. LCTMS showed push and pull as the primary factors as well as the six sub-factors, and LCTCS only showed constraint as the main factor as well as the four sub-factors. The fit indices for all four models are separately presented in Table 8.

The CFI, SRMR, and RMSEA indicated adequate data to model fit, for all models based on the general guidelines. To compare the first-order and second-order models, the researcher examined the NCP, AIC, and ECVI. Although the first-order models were marginally better, the second-order models were both reasonably close and significantly easier to use in research and practice. Pan et al. [65] pointed out that when the models are all adapted, the higher-order model has the principle of simplification. Therefore, this study chooses the second-order models as the best hierarchical model for LCTMS (Figure 2) and LCTCS (Figure 3).

### 4.3. Discussion

This study surveys independent tourists from low-carbon travel and non-low-carbon travel sites and evaluates the reliability and validity of two scales. The results exhibit high goodness of fit, supporting the second-order two-factor structure of low-carbon travel motivation with six potential constructs and the second-order single-factor form of low-carbon travel constraint with four possible constructs. The CR of each construct is at least 0.7, indicating that the scale reliability is acceptable. This study ensures that each scale has reasonable validity through construction, convergence, differential, and criterion-related validity. The results indicate a need to explore the essential constructs and contents of low-carbon travel motivation and low-carbon travel constraint for independent tourists and tour operators, to confirm why tourists do or do not engage in low-carbon travel.

Scholars have been increasingly interested in determining the motivations and constraints behind tourists’ decisions to engage in low-carbon travel [5,8]. This study’s results reflect the motivations, constraints, and ESCR behaviors of tourists that have been proposed in the other low-carbon travel literature. Kuo and Dai [5] and Horng et al. [8], for example, have discussed environmental protection motivations, while Dickinson et al. [44] and Dällenbach [42] discussed intrapersonal constraints, interpersonal constraints, and structural constraints.

The relevant literature has stressed that low-carbon travel contributes to sustainable tourism development [5,8]—Horng et al. [54] drew a low-carbon travel literacy scale from the perspectives of tourism practitioners. Hsiao [68] contributed a low-carbon tourism evaluation index system for travel agencies. Lee and Jan [52] developed a low-carbon tourism experience scale to guide tourists and managers to reduce carbon emissions. Tsaur and Dai [69] provided a low-carbon travel scale for independent tourists to engage in a low-carbon travel. However, Horng et al. [54] only adopted potential literacy from some internal motivations. Hsiao [68] focused on establishing the industrial-level low-carbon tourism index. Lee and Jan [52] and Tsaur and Dai [69] all encouraged studies to focus on examining the causal structural models of the low-carbon tourism experience and low-carbon travel behavior.

Since few studies have comprehensively examined the low-carbon travel motivation and low-carbon travel constraint in the tourism industry, this study is based on a push and pull structure of tourism motives as well as a three-dimensional framework for constraints and, then, emphasizes the diversified orientation of low-carbon travel motivation and low-carbon travel constraint. This study suggested various critical factors indicators gathered from in-depth interviews and questionnaires. Furthermore, there are still many studies that continued to explore the relationships between travel motivation, constraint, and experience. Compared to past studies, this study provided the necessary concept and measurements of tourists’ low-carbon travel motivation and constraint in the low-carbon tourism context; the LCTMS and LCTCS developed in this article highlight the independent tourists’ perspectives.

## 5. Conclusions

Based on the scale development process recommended by Churchill [49], this study integrates qualitative and quantitative methods to develop two measurement scales. Tourists’ diverse views of questionnaires were collected from the literature review, interviews, content analysis, and expert discussion. According to the scale development guidelines provided by Churchill [49], Ager [48], and Bhatt et al. [50], this study constructs a motivation scale and a constraint scale suitable for explaining tourists’ reasons for engaging in low-carbon travel. After a rigorous scale development process, this study establishes essential constructs for low-carbon travel motivations and constraints.

Low-carbon travel is a broad concept, and independent tourists’ decisions on it vary according to their different motivations and constraints. The researchers believe that an in-depth understanding of motivations and constraints can help tourism operators encourage independent tourists to overcome those constraints and engage in low-carbon travel. Thus, the research results reveal the primary low-carbon travel motivations and low-carbon travel constraints of independent tourists, regardless of a tourist’s travel purpose or mode.

The practical contributions of this study are (1) helping the tourism industry identify why independent tourists may or may not engage in low-carbon travel, (2) enabling the industry to provide the services that tourists want and need, and (3) assisting the industry in planning its corresponding marketing strategy. Theoretically, this study integrates the different motivations and constraints of low-carbon travel and lays out a reasonable theoretical foundation. The construction of the LCTMS and LCTCS adds to previous knowledge as well as provides a practical assessment and guidance tool for tourists and scholars. The two scales are appropriate tools for low-carbon travel market segmentation.

### 5.1. Theoretical Implications

This study confirms the stability of the LCTMS and LCTCS and the structure, reliability, and validity of the low-carbon travel behavior scale. Therefore, the LCTMS and LCTCS can, indeed, provide low-carbon travel operators and authorities with why tourists choose to engage or not engage in low-carbon travel and, then, motivate tourists to overcome constraints and practice low-carbon travel. Concerning the spirit of theoretical construction, the significance of using this research scale is to understand the psychological and behavioral context of independent tourists engaged in low-carbon travel.

From theoretical development, researchers have focused on the theory of planned behavior [5] and the theory of protection motivation [8] to explore the antecedents of tourists’ low-carbon travel intentions. However, now the LCTMS and LCTCS can be used to measure why tourists make low-carbon travel decisions. When motivations and constraints conflict, constraint negotiation will enable tourists to adjust. Suppose researchers further explore the negotiation constraints in tourists’ low-carbon travel decision-making and link low-carbon travel intentions and behaviors. In that case, they may additionally construct a causal model to interpret low-carbon tourists’ behavior. This research scale can be used to explore the effects of low-carbon motivation and constraint on constraint negotiation and, further, explore the influence of constraints on travel behavior in different tourism situations.

### 5.2. Management Implications

Tourists and tourism authorities can further apply the content of these two scales, integrate recreational resources, policy dissemination, media, and, even, knowledge acquisition, and strengthen independent tourists’ low-carbon travel motivations and reduce their constraints. Thus, this study has several implications for management.

#### 5.2.1. Low-Carbon Travel Motivation

Low-carbon travel differs from other types of tourism, in that it emphasizes ESCR and lessens the adverse effects on the environment. The competent authorities of tourism destinations should actively educate independent tourists about environmental protection. As independent tourists increasingly favor ESCR, reduce their resources waste, and lessen the harmful effects of tourism on the environment, their motivation to engage in low-carbon travel will increase. Second, the competent authorities of tourism destinations should improve low-carbon travel services and the environment to provide independent tourists with more opportunities to explore the culture of tourism destinations. This approach makes it possible for independent tourists to understand the difference between low-carbon travel and general tourism.

Third, the competent authorities of tourism destinations should emphasize the escape and social interaction aspects of low-carbon travel. So, independent tourists can also escape from their daily life or work, enjoy physical and mental relaxation, spend time with relatives and friends, and, even, meet people with the same hobbies or interests. The tourism industry’s environmental protection appeals and measures are a fourth powerful motivation. The tourism industry should apply different environmental protection labels according to their current conditions or ability to improve, propose specific environmental protection slogans and demands, and provide ESCR facilities through multiple avenues. The above measures will help independent tourists better understand where to choose low-carbon operators.

Lin [68] discovered a significant canonical correlation between purchasing motivations and product attributes, by investigating the food-souvenir-purchasing behaviors of Chinese tourists. The competent authorities of tourism destinations should encourage residents to produce local products and food as well as cooperate with government agencies to promote local products and food to independent tourists. Finally, tourism destinations ought to provide convenient public transportation, such as trains, buses, and high-speed rail, as well as well-planned bicycle lanes and hardware facilities for landscape trails, encouraging independent tourists to choose a low-carbon and healthy way to move between attractions.

#### 5.2.2. Low-Carbon Travel Constraint

Nowadays, in contrast to group tourists, independent tourists make their arrangements and consider the constraints of many ESCR measures in low-carbon travel. The tourism industry and authorities should provide independent tourists a convenient ESCR and comfortable tourism situation, along with much low-carbon travel information to improve tourist views of low-carbon travel. For example, Horng et al. [3] believed that transportation options at tourism destinations should be more convenient, to encourage tourists to travel by mass transit, boosting passenger load capacity and reducing per visitor carbon footprint. Besides, tourism operators and low-carbon travel authorities must work together closely to provide more low-carbon travel-related products, reducing low-carbon travel prices and increasing the attractiveness of low-carbon travel options [11].

Independent tourists wanting to engage in low-carbon travel face more intrapersonal constraints than group-package tourists. Since time and budget planning are more casual for independent tourists than group-package tourists, the tourism industry and competent authorities of low-carbon travel destinations should provide sufficient knowledge to tourists and plan marketing programs to attract independent tourists by giving incentives such as time, finance, and a low-carbon journey.

Third, the competent authorities of tourism destinations should strive to promote low-carbon travel and encourage independent tourists to adopt low-carbon travel to develop a thriving low-carbon travel market. The tourism industry and the authorities of low-carbon travel could encourage independent tourists to travel with family and friends, by offering discounts and actively communicating that low-carbon travel can strengthen emotional connections with family and friends.

### 5.3. Limitations and Suggestions for Further Research

First, the two scales constructed in this study are designed from the perspective of independent tourists. They may not suit group tourists. Since tourists primarily seek convenience and comfort, it is impossible to comprehensively consider the practice of low-carbon energy conservation, to explore the motivations and constraints of low-carbon travel from the perspective of group tourists. Further studies may adopt hieratic analysis to explore individual tourists’ low-carbon travel motivations and constraints in group tours.

Second, further research can be conducted on low-carbon travel and non-low-carbon travel sites in other regions or countries, covering tourists with and without low-carbon travel experiences to test the external validity of these scales. Third, researchers can use the two scales to further examine the relationships between tourists’ low-carbon travel motivations and low-carbon travel constraints and other variables (such as attitudes, subjective norms, constraint negotiation, and perceived behavioral control) to strengthen the theoretical basis of relevant research, such as the theories of planned behavior [5] and protection motivation [8].

Finally, further research can use cluster analysis and spectral concepts to study high-carbon tourism, moderate and mild low-carbon travel, and non-low-carbon travel experiences, comparing the low-carbon travel motivations and low-carbon travel constraints of tourists’ groups with the different levels of the low-carbon travel experience.

## Figures and Tables

**Figure 1 ijerph-19-05123-f001:**
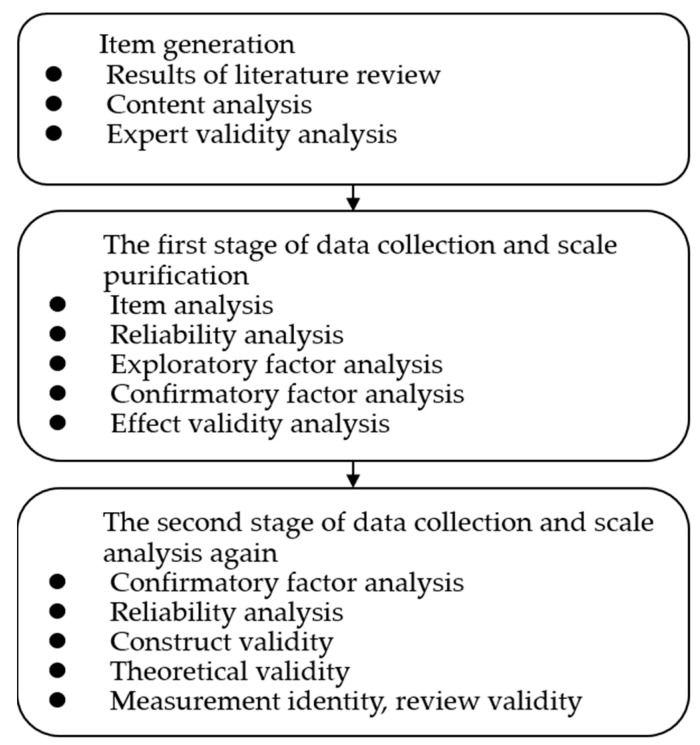
Scale development process.

**Figure 2 ijerph-19-05123-f002:**
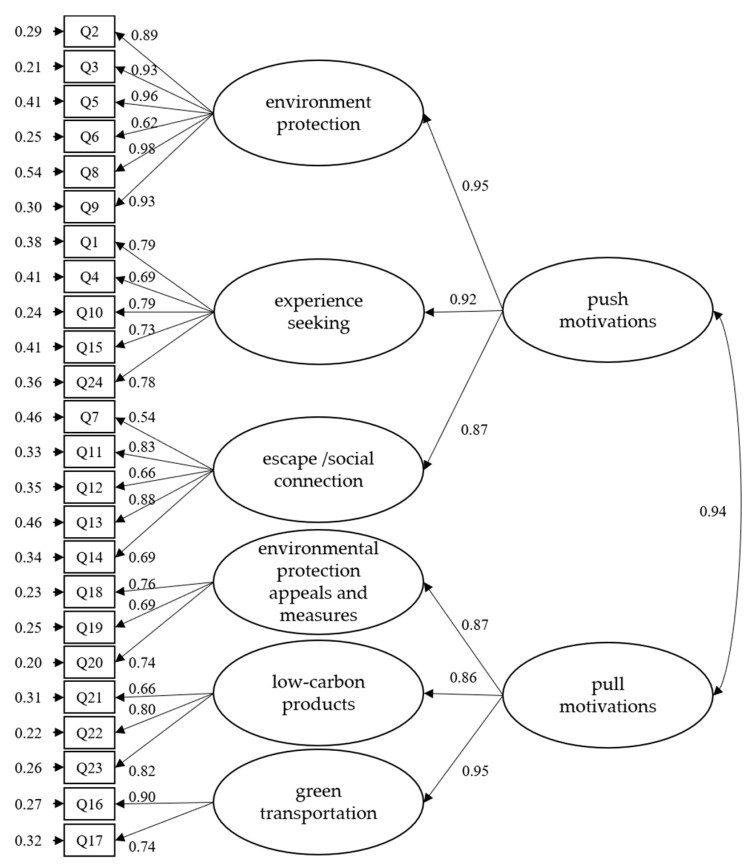
The second-order model of LCTMS.

**Figure 3 ijerph-19-05123-f003:**
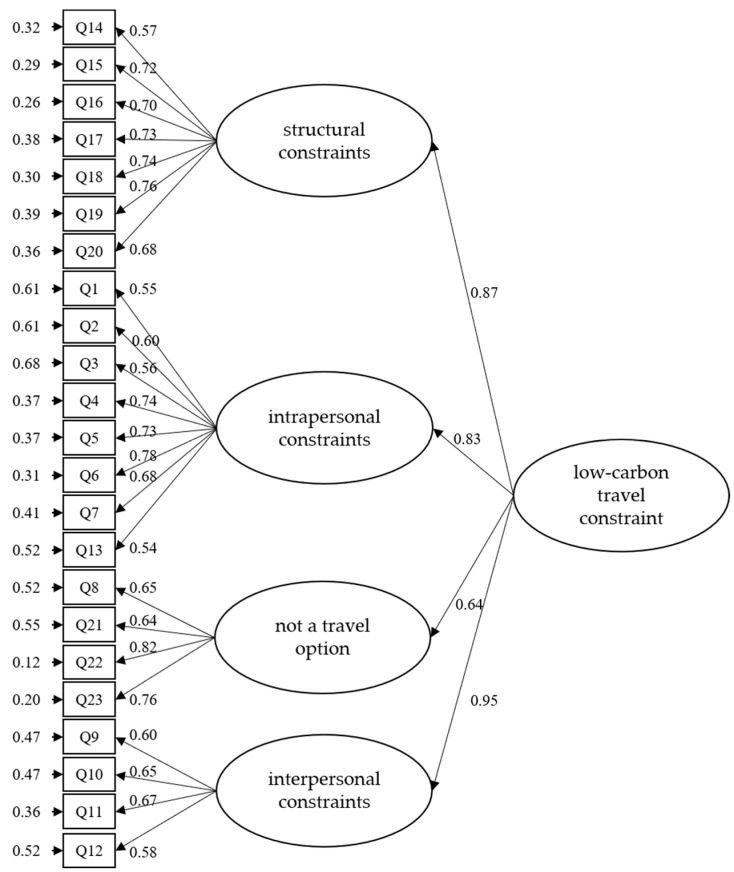
The second-order model of LCTCS.

**Table 1 ijerph-19-05123-t001:** The first sample’s profile and travel characteristics—from low-carbon destinations (*n* = 382).

Variable	Items	Number (%)	Variable	Items	Number (%)
Gender	Male	179 (46.9%)	Career	Agriculture, forestry, animal husbandry	9 (2.4%)
	Female	203 (53.1%)		Miner manufacturing	22 (5.8%)
Age	Under 20 years old	72 (18.8%)		Traditional industry, social service industry	80 (20.9%)
	21–30 years old	166 (43.5%)		Education, information, government-industry	41 (10.7%)
	31–40 years old	57 (14.9%)		Student, family management, retirement	230 (60.2%)
	41–50 years old	44 (11.5%)	Experiences of low-carbon travel in the past year (including this time)	1 time	265 (69.4%)
	51–60 years old	30 (7.9%)			
	More than 60 years old	13 (3.4%)		2 times	65 (17.0%)
Education level	Under senior high school	13 (3.4%)		3 times	23 (6.0%)
	Senior high school	55 (14.4%)		more than 3 times	29 (7.6%)
	College/University	255 (66.8%)			
	Master/Doctor	59 (15.4%)	Days of this tour	1 day	92 (24.1%)
Marital status	Single	250 (65.4%)		2 days	97 (25.4%)
	Married	111 (29.1%)		3 days	116 (30.4%)
	Other	21 (5.6%)		More than 3 days	77 (20.2%)
Monthly income (RMB)	Less than 5000	189 (49.5%)			
	5001–10,000	115 (30.1%)			
	10,001–15,000	43 (11.3%)			
	More than 15,000	35 (9.2%)			

**Table 2 ijerph-19-05123-t002:** EFA results of LCTMS—sample from low-carbon destination (*n* = 382).

Factors/Items	Factor Loading	Eigenvalue	Cumulative Variation	Cronbach’s α
I am engaged in low-carbon travel because of...				
Factor 1: environment protection		5.02	20.90%	0.88
3. I identify the concept of ESCR.	0.83			
2. it reduces the impact on the tourism environment.	0.76			
8. I respect nature.	0.71			
5. it can reduce resource waste.	0.70			
6. I am interested in energy saving and carbon reduction.	0.70			
9. I am responsible for the environment.	0.69			
Factor 2: experience seeking		2.81	32.61%	0.80
1. it helps my health.	0.63			
10. it provides opportunities for family/parental environmental education.	0.59			
4. I can get low-carbon tourism knowledge.	0.54			
15. I can deeply explore the humanities of travel destinations in a fixed place.	0.52			
24. I can experience the difference between low-carbon tourism and general tourism	0.50			
Factor 3: escape/social connection		2.68	43.78%	0.83
11. I can stay away from daily life/work environment.	0.84			
12. I can avoid daily life/work stress.	0.82			
7. I can know a person having the same habit in ESCR.	0.73			
13. I can get peace of mind and body.	0.68			
14. I can contact my relatives and friends.	0.57			
Factor 4: environmental protection appeals and measures		2.47	54.09%	0.87
19. environmental slogans and appeals of travel industries.	0.84			
18. the industries have environmental labels (such as environmentally friendly hotels, restaurants, and environmentally friendly vehicles).	0.77			
20. travel industries provide ESCR measures.	0.76			
Factor 5: low-carbon products		1.95	62.21%	0.81
22. I can purchase local products.	0.78			
23. I can eat local foods.	0.72			
21. I can buy environmentally friendly products.	0.62			
Factor 6: green transportation		1.76	69.56%	0.73
17. perfect bicycle or trail facilities.	0.68			
16. convenient mass transportation (train, bus, high-speed rail).	0.66			

**Table 3 ijerph-19-05123-t003:** EFA results of LCTCS—sample from low-carbon destination (*n* = 382).

Factors/Items	Factor Loading	Eigenvalue	Cumulative Variation	Cronbach’s α
I don’t engage in low-carbon travel because of...				
Factor 1: structural constraints		4.19	18.22%	0.84
17. there is insufficient information on low-carbon destinations.	0.76			
19. fewer operators are offering low-carbon tourism.	0.75			
15. low-carbon tourism is not convenient enough for me.	0.74			
18. less low-carbon tourism products.	0.73			
16. low carbon travel is not comfortable enough for me.	0.68			
20. low-carbon tourism content is not attractive.	0.59			
14. the price of low-carbon tourism is unreasonable for me.	0.53			
Factor 2: intrapersonal constraints		3.43	33.12%	0.85
6. I have no energy for low-carbon tourism.	0.77			
4. I have no time for low-carbon tourism.	0.76			
5. I have no money for low-carbon tourism.	0.72			
3. my physical condition is not suitable for low-carbon tourism.	0.70			
2. I don’t know how to engage in low-carbon tourism.	0.65			
1. my low-carbon tourism knowledge is not enough.	0.64			
13. I have many family obligations.	0.53			
7. I have a lot of work or scholarship responsibilities.	0.50			
Factor 3: not a travel option		3.36	47.75%	0.82
23. low-carbon life is not what I want.	0.80			
22. low-carbon tourism has never been my travel option.	0.78			
8. I am not interested in low-carbon tourism.	0.60			
21. I have many other travel alternatives.	0.55			
Factor 4: interpersonal constraints		2.85	60.12%	0.72
12. my family/friends are not interested in low-carbon tourism.	0.72			
9. I lack partners.	0.66			
10. I must consider the physical condition of my partners.	0.63			
11. my family/friends are not interested in low-carbon tourism.	0.55			

**Table 4 ijerph-19-05123-t004:** The second sample’s profile and travel characteristics—from non-low-carbon destinations (*n* = 390).

Variable	Items	Number (%)	Variable	Items	Number (%)
Gender	Male	150 (38.5%)	Career	Agriculture, forestry, animal husbandry	9 (2.3%)
	Female	240 (61.5%)		Miner manufacturing	12 (3.1%)
Age	Under 20 years old	167 (42.8%)		Traditional industry, social service industry	59 (15.1%)
	21–30 years old	147 (37.7%)		Education, information, government-industry	33 (8.5%)
	31–40 years old	33 (8.5%)		Student, family management, retirement	277 (71.0%)
	41–50 years old	9 (2.3%)			
	51–60 years old	21 (5.4%)			
	More than 60 years old	13 (3.3%)	Experiences of low-carbon travel in the past year (including this time)	never	196 (5.3%)
Education level	Under senior high school	9 (2.3%)		1 time	92 (23.6%)
	Senior high school	29 (7.4%)		2 times	62 (15.9%)
	College/University	321 (82.3%)		3 times	18 (4.6%)
	Master/Doctor	31 (8.0%)		More than 3 times	22 (5.6%)
Marital status	Single	314 (80.5%)	Days of this tour	1 day	173 (44.4%)
	Married	60 (15.4%)		2 days	116 (29.7%)
	Other	16 (4.1%)		3 days	48 (12.3%)
Monthly income (RMB)	Less than 5000	272 (69.7%)		More than 3 days	53 (13.6%)
	5001–10,000	83 (21.3%)			
	10,001–15,000	21 (5.4%)			
	More than 15,000	14 (3.6%)			

**Table 5 ijerph-19-05123-t005:** CFA results of LCTMS—sample from non-low-carbon destinations (*n* = 390).

Factors/Items	SFL	t-Value	SMC	CR	AVE
I am engaged in low-carbon travel because of...					
Factor 1: environment protect				0.94	0.72
3. I identify the concept of ESCR.	0.89	22.75	0.81		
5. it can reduce resource waste.	0.89	22.39	0.79		
9. I am responsible for the environment.	0.88	22.32	0.78		
8. I respect nature.	0.87	21.90	0.76		
2. it reduces the impact on the tourism environment.	0.86	21.17	0.73		
6. I am interested in energy saving and carbon reduction.	0.68	15.10	0.54		
Factor 2: experience seeking				0.88	0.61
10. it provides opportunities for family/parental environmental education.	0.81	19.28	0.66		
1. it helps my health.	0.80	18.82	0.63		
24. I can experience the difference between low-carbon tourism and general tourism	0.79	18.59	0.64		
15. I can deeply explore the humanities of travel destinations in a fixed place.	0.76	17.52	0.57		
4. I can get low-carbon tourism knowledge.	0.74	16.76	0.54		
Factor 3: escape/social connection				0.87	0.57
13. I can get peace of mind and body.	0.87	21.17	0.76		
11. I can stay away from daily life/work environment.	0.82	19.12	0.67		
14. I can contact my relatives and friends.	0.77	17.39	0.59		
12. I can avoid daily life/work stress.	0.71	15.49	0.50		
7. I can know a person having the same habit in ESCR.	0.59	12.22	0.59		
Factor 4: environmental protection appeals and measures				0.88	0.70
18. the industries have environmental labels (such as environmentally friendly hotels, restaurants, and environmentally friendly vehicles).	0.85	20.42	0.73		
20. travel industries provide ESCR measures.	0.85	20.45	0.73		
19. environmental slogans and appeals of travel industries.	0.81	18.85	0.66		
Factor 5: low-carbon products				0.86	0.68
22. I can purchase local products.	0.86	20.76	0.74		
23. I can eat local foods.	0.85	20.11	0.73		
21. I can buy environmentally friendly products.	0.77	17.47	0.59		
Factor 6: green transportation				0.82	0.70
16. convenient mass transportation (train, bus, high-speed rail).	0.86	20.64	0.75		
17. perfect bicycle or trail facilities.	0.81	18.74	0.65		

Note: SFL is standard factor loading, SMC is squared multiple correlations, CR is composite reliability, and AVE is average variation extracted.

**Table 6 ijerph-19-05123-t006:** CFA results of LCTCS—sample from non-low-carbon destinations (*n* = 390).

Factors/Items	SFL	t-Value	SMC	CR	AVE
I can’t engage in low-carbon travel because of...					
Factor 1: structural constraints				0.93	0.60
15. low-carbon tourism is not convenient enough for me.	0.81	18.80	0.65		
16. low carbon travel is not comfortable enough for me.	0.81	18.86	0.65		
18. less low-carbon tourism products.	0.81	18.73	0.65		
17. there is insufficient information on low-carbon destinations.	0.77	17.34	0.58		
19. fewer operators are offering low-carbon tourism.	0.77	17.51	0.59		
20. low-carbon tourism content is not attractive.	0.76	16.93	0.56		
14. the price of low-carbon tourism is unreasonable for me.	0.72	15.78	0.51		
Factor 2: intrapersonal constraints				0.89	0.50
6. I have no energy for low-carbon tourism.	0.82	18.93	0.67		
4. I have no time for low-carbon tourism.	0.78	17.54	0.60		
5. I have no money for low-carbon tourism.	0.77	17.27	0.59		
7. I have a lot of work or scholarship responsibilities.	0.73	16.00	0.53		
2. I don’t know how to engage in low-carbon tourism.	0.62	12.70	0.69		
13. I have many family obligations.	0.61	12.40	0.56		
1. my low-carbon tourism knowledge is not enough.	0.60	11.79	0.57		
3. my physical condition is not suitable for low-carbon tourism.	0.60	11.65	0.51		
Factor 3: not a travel option				0.86	0.68
22. low-carbon tourism has never been my travel option.	0.92	22.69	0.85		
23. low-carbon life is not what I want.	0.87	20.56	0.75		
8. I am not interested in low-carbon tourism.	0.67	14.28	0.64		
21. I have many other travel alternatives.	0.66	14.01	0.53		
Factor 4: interpersonal constraints				0.81	0.50
11. my family/friends are not interested in low-carbon tourism.	0.75	16.01	0.55		
10. I must consider the physical condition of my partners.	0.70	14.79	0.58		
9. I lack partners.	0.67	13.74	0.55		
12. my family/friends are not interested in low-carbon tourism.	0.61	12.40	0.50		

Note: SFL is standard factor loading, SMC is squared multiple correlations, CR is composite reliability, and AVE is average variation extracted.

**Table 7 ijerph-19-05123-t007:** Discriminatory validity of Low-Carbon Motivation and Constraint Scale.

Constructs	1	2	3	4	5	6	7	8	9	10
1. environmental protection	**0.85**									
2. experience seeking	0.76	**0.78**								
3. escape/social connection	0.70	0.75	**0.76**							
4. environmental protection appeals and measures	0.62	0.66	0.56	**0.84**						
5. low-carbon products	0.63	0.70	0.61	0.65	**0.82**					
6. green transportation	0.69	0.75	0.62	0.67	0.63	**0.84**				
7. structural constraints							**0.77**			
8. intrapersonal constraints							0.68	**0.71**		
9. not a travel option							0.65	0.61	**0.79**	
10. interpersonal constraints							0.59	0.56	0.55	**0.71**

Note: the bold figure is the square root of the average variation extract for each factor.

**Table 8 ijerph-19-05123-t008:** Comparison of competitive models.

	Low-Carbon Travel Motivation Scale		Low-Carbon Travel Constraint Scale	
Indexes	The First-Order Model with Six Factors	The Second-Order Model with a Push-and-Pull Factor and Six Sub-Factors	The First-Order Model with Four Factors	The Second-Order Model with a Constraint Factor and Four Sub-Factors
absolute fit indices				
χ^2^	776.36	799.81	661.38	664.98
df	237	245	224	226
χ^2^/df	3.28	3.26	2.95	2.94
GFI	0.86	0.85	0.88	0.88
AGFI	0.82	0.82	0.86	0.86
SRMR	0.05	0.05	0.08	0.08
RMSEA	0.076	0.076	0.078	0.078
relative fit indices				
NFI	0.98	0.98	0.93	0.93
NNFI	0.98	0.98	0.93	0.93
CFI	0.98	0.98	0.94	0.94
IFI	0.98	0.98	0.94	0.94
RFI	0.97	0.97	0.92	0.92
parsimony fit indices				
NCP	539.36	554.81	437.38	438.98
AIC	902.36	909.81	765.38	764.98
ECVI	2.33	2.35	4.55	4.55

## Data Availability

Not applicable.

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
