# Peer review of "Low-Carbon Travel Motivation and Constraint: Scales Development and Validation"

_ijerph, 2022, doi:10.3390/ijerph19095123_

Round 1

Reviewer 1 Report

The article entitled: ‘Low-carbon travel motivation and constraint: Scales development and validation’, is the very interesting research report. It rises up important topic which is not much recognized yet. The topic of low-carbon traveling motivations is worth to survey from many perspectives.

The topic is well explained and based on the previous researches.

However I have a remark to the way the article is written. In my opinion the paper is a little too long without the reason for that. It contains a lot of repetitions. That is why it is hard to follow all the research procedure and results. The introduction to the topic is too long. The ‘Introduction’ and ‘Literature review’ parts are almost about the same, they could be combined into one. The ‘Discussion’ part doesn't discuss the results compering to other similar researches. ‘Conclusion’, more than half of it is a summary of all the article. Only points 5.2 and 5.3 conclude the results but could be more contracted.

Author Response

Response Letter

Journal Title: International Journal of Environmental Research and Public Health

Manuscript Reference No. 1661699

Manuscript Title: Low-carbon travel motivation and constraint: Scales development and validation

We sincerely appreciate all the detailed comments and suggestions made by the reviewers. All comments are considered carefully and summarized as follows, and all the revisions are marked in Red in the original paper for reviewers’ readability. In addition, the revised manuscript has been sent to Wallace Language Editing Services for cosmetic edit. We believe the presentation of this revised manuscript should be more expressive and more readable than the original manuscript. Please do not hesitate to give us any comments and we also look forward to your further decision.

Responses from the authors to Reviewer 1:

Review Report (Reviewer 1)

Comments and Suggestions for Authors

The article entitled: ‘Low-carbon travel motivation and constraint: Scales development and validation’, is the very interesting research report. It rises up important topic which is not much recognized yet. The topic of low-carbon traveling motivations is worth to survey from many perspectives. The topic is well explained and based on the previous researches.

Response: Many thanks for your comments. It’s our honor to receive your regard as positive.

However I have a remark to the way the article is written. In my opinion the paper is a little too long without the reason for that. It contains a lot of repetitions. That is why it is hard to follow all the research procedure and results. The introduction to the topic is too long. The ‘Introduction’ and ‘Literature review’ parts are almost about the same, they could be combined into one.

Response: We agree your opinion. To reduce this article’s length, we did the follow modifications:

First, we revised this submission by following the MDPI guideline. The citations that 3 or more authors are marked red color as “first author, et al.” in the whole manuscript.

Second, we deleted and modified some paragraphs in section “Introduction” and “Literature review”. The modified paragraphs are marked red colors.

Third, we deleted and changed some inappropriate references.

Fourth, we improved the English again.

By these way, the total words are reduced from 13468 to 12590.

The ‘Discussion’ part doesn't discuss the results compering to other similar researches.

Response: Thanks for your conclusion. We have added similar research to compare to. Please see line 660 to 686.

Scholars have been increasingly interested in determining the motivations and constraints behind tourists’ decisions to engage in low-carbon travel [5,8]. This study’s results reflect the motivations, constraints, and ESCR behaviors of tourists that have been proposed in other low-carbon travel literature. Kuo and Dai [5] and Horng, et al. [8], for example, have discussed environmental protection motivations, while Dickinson, et al. [44] and Dällenbach [42] discussed intrapersonal constraints, interpersonal constraints, and structural constraints.

Relevant literature has stressed that low-carbon travel contributes to sustainable tourism development [5,8]—Horng, et al. [54] drew a low-carbon travel literacy scale from the perspectives of tourism practitioners. Hsiao [68] contributed a low-carbon tourism evaluation index system for travel agencies. Lee and Jan [52] developed a low-carbon tourism experience scale to guide tourists and managers to reduce carbon emissions. However, Horng, et al. [54] only adopt potential literacy from some internal motivations. Hsiao [68] focused on establishing the industrial-level low-carbon tourism index. Lee and Jan [52] encouraged studies o focus on examing the causal structural models of low-carbon tourism experience.

Since few studies have comprehensively examined the low-carbon travel motivation and low-carbon travel constraint in the tourism industry, this study is based on a push and pull structure of tourism motives and a three-dimensional framework for constraints and then emphasizes the diversified orientation of low-carbon travel motivation and low-carbon travel constraint. This study suggested various critical factors indicators gathered from in-depth interviews and questionnaires. Furthermore, there are still many studies that continued to explore the relationships between travel motivation, constraint, and experience. Compared to past studies, this study provided the necessary concept and measurements of tourists’ low-carbon travel motivation and constraint in the low-carbon tourism context, the LCTMS and LCTCS developed in this article highlight the independent tourists' perspectives.

Added references:

Lee, T.H.; Jan, F.-H. The low-carbon tourism experience: A multidimensional scale development. Journal of Hospitality & Tourism Research 2019, 43, 890-918, doi:10.1177/1096348019849675.

Hsiao, T.-Y. Developing a dual-perspective low-carbon tourism evaluation index system for travel agencies. J. Sustainable Tour. 2016, 24, 1604-1623, doi:10.1080/09669582.2015.1136633.

‘Conclusion’, more than half of it is a summary of all the article. Only points 5.2 and 5.3 conclude the results but could be more contracted.

Response: We agree your comments. Hence, we contracted the section 5.2 and 5.3 to make them more readable. Please see pp. 22-23.

5.2.1. Low-Carbon Travel Motivation

Low-carbon travel differs from other types of tourism in that it emphasizes ESCR and lessens the adverse effects on the environment. The competent authorities of tourism destinations should actively educate independent tourists about environmental protection. As independent tourists increasingly favor ESCR, reduce their resources waste, and lessen the harmful effects of tourism on the environment, their motivation to engage in low-carbon travel will increase. Second, the competent authorities of tourism destinations should improve low-carbon travel services and the environment to provide independent tourists with more opportunities to explore the culture of tourism destinations. This approach makes it possible for independent tourists to understand the difference between low-carbon travel and general tourism.

Third, the competent authorities of tourism destinations should emphasize the escape and social interaction aspects of low-carbon travel. So that independent tourists also can escape from their daily life or work, enjoy physical and mental relaxation, spend time with relatives and friends, and even meet people with the same hobbies or interests. The tourism industry's environmental protection appeals and measures are a fourth powerful motivation. The tourism industry should apply different environmental protection labels according to their current conditions or ability to improve, propose specific environmental protection slogans and demands, and provide ESCR facilities through multiple avenues. The above measures will help independent tourists better understand where to choose low-carbon operators.

Lin [68] discovered a significant canonical correlation between purchasing motivations and product attributes by investigating the food souvenir-purchasing behaviors of Chinese tourists. The competent authorities of tourism destinations should encourage residents to produce local products and food and cooperate with government agencies to promote local products and food to independent tourists. Finally, tourism destinations ought to provide convenient public transportation, such as trains, buses, and high-speed rail, and well-planned bicycle lanes and hardware facilities for landscape trails, encouraging independent tourists to choose a low-carbon and healthy way to move between attractions.

5.2.2. Low-Carbon Travel Constraint

Nowadays, in contrast to group tourists, independent tourists make their arrangements and consider the constraints of many ESCR measures in low-carbon travel. The tourism industry and authorities should provide independent tourists a convenient, ESCR, and comfortable tourism situation, along with much low-carbon travel information to improve tourist views of low-carbon travel. For example, Horng, et al. [3] believed that transportation options at tourism destinations should be more convenient to encourage tourists to travel by mass transit, boosting passenger load capacity and reducing per visitor carbon footprint. Besides, tourism operators and low-carbon travel authorities must work together closely to provide more low-carbon travel-related products, reducing low-carbon travel prices and increasing the attractiveness of low-carbon travel options [11].

Independent tourists wanting to engage in low-carbon travel face more intrapersonal constraints than group-package tourists. Since time and budget planning are more casual for independent tourists than group-package tourists, the tourism industry and competent authorities of low-carbon travel destinations should provide sufficient knowledge to tourists and plan marketing programs to attract independent tourists by giving incentives such as time, finance, and a low-carbon journey.

Third, the competent authorities of tourism destinations should strive to promote low-carbon travel and encourage independent tourists to adopt low-carbon travel to develop a thriving low-carbon travel market. The tourism industry and the authorities of low-carbon travel could encourage independent tourists to travel with family and friends by offering discounts and actively communicating that low-carbon travel can strengthen emotional connections with family and friends.

5.3.      Limitations and suggestions for further research

First, the two scales constructed in this study are designed from the perspective of independent tourists. They may not suit group tourists. Since tourists primarily seek convenience and comfort, it is impossible to comprehensively consider the practice of low-carbon energy conservation to explore the motivations and constraints of low-carbon travel from the perspective of group tourists. Further studies may adopt hieratic analysis to explore individual tourists' low-carbon travel motivations and constraints in group tours.

Second, further research can be conducted on low-carbon travel and non- low-carbon travel sites in other regions or countries, covering tourists with and without low-carbon travel experiences to test the external validity of these scales. Third, researchers can use the two scales to examine further the relationships between tourists' low-carbon travel motivations and low-carbon travel constraints and other variables (such as attitudes, subjective norms, constraint negotiation, and perceived behavioral control) to strengthen the theoretical basis of relevant research, such as the theories of planned behavior [5] and protection motivation [8].

Finally, further research can use cluster analysis and spectral concepts to study high-carbon tourism, moderate and mild low-carbon travel, and non- low-carbon travel experiences and compare the low-carbon travel motivations and low-carbon travel constraints of tourists’ groups with different levels of low-carbon travel experience.

Reviewer 2 Report

I appreciate the authors pursing a very timely and important topic. The authors should be applauded for the thorough literature review and data collection. I have two main concerns.

  1. It’s not clear how the factors of Low-Carbon Travel Motivation were identified. The authors wrote “The push and pull factors of Michael, Nyadzayo, Michael and Balasubramanian [4] play as the main theoretical framework in this study.” However, this study is not about low carbon travel motivation and there are new factors in the manuscript. The authors should provide clear theoretical reasonings behind it.
  2. What I see Low-Carbon Travel Constraint scale is the mere inclusion of the word “low-carbon.” All items already existed and the word was only changed. If there is no specific construct associated with low-carbon travel constraint, and all we do is to add the context, I don’t see much value in the developed scale. This comment goes back to my first question about how the authors identified those factors. I believe interviews with consumers should have been done to identify the low-carbon context-specific factors, instead of adopting previous factors, as this is a new study context. The authors should clear this up by explaining why the new scales differ from other and how the new scales provide value to researchers and practitioners.

Other issues:

  1. I suggest providing a figure of the both second-order scales indicating those are the final models.
  2. I think the authors can further make the most of the data. Motivation and constraint have been often studied in a single model. The authors can further test the relationship between the two constructs as evidence of predictive validity.
  3. Were participants aware of what ESCR stood for? Abbreviation in the item is not a good practice, even though there was an explanation, as some people might not pay attention. The authors might want to spell out.

Author Response

Response Letter

Journal Title: International Journal of Environmental Research and Public Health

Manuscript Reference No. 1661699

Manuscript Title: Low-carbon travel motivation and constraint: Scales development and validation

We sincerely appreciate all the detailed comments and suggestions made by the reviewers. All comments are considered carefully and summarized as follows, and all the revisions are marked in Red in the original paper for reviewers’ readability. In addition, the revised manuscript has been sent to Wallace Language Editing Services for cosmetic edit. We believe the presentation of this revised manuscript should be more expressive and more readable than the original manuscript. Please do not hesitate to give us any comments and we also look forward to your further decision.

Responses from the authors to Reviewer 2:

Review Report (Reviewer 2)

Comments and Suggestions for Authors

I appreciate the authors pursing a very timely and important topic. The authors should be applauded for the thorough literature review and data collection.

Response: It’s our honor to receive your regard as positive.

I have two main concerns.

  1. It’s not clear how the factors of Low-Carbon Travel Motivation were identified. The authors wrote “The push and pull factors of Michael, Nyadzayo, Michael and Balasubramanian [4] play as the main theoretical framework in this study.” However, this study is not about low carbon travel motivation and there are new factors in the manuscript. The authors should provide clear theoretical reasonings behind it.

Response: Thanks for your comment. We revised our theoretical base from Michael, Nyadzayo, Michael and Balasubramanian to Crompton (1979), which is the most classical reference of push and pull motivations. Please see line 104.

Although these researchers have mainly used factor analysis or cluster analysis to determine the various travel motivations of their subjects, since the nature of travel is a series of travel activities to satisfy people’s inner social-psychology needs or the external cultural seeking of destination, tourist motivation has tended to revolve around the concepts of "pull" and "push" [20]. Most discussions in tourism have applied the theory of push-and-pull motivation when explaining why people travel [21-24]. Therefore, the push and pull factors of Crompton [20] play the main theoretical framework in this study.

  1. What I see Low-Carbon Travel Constraint scale is the mere inclusion of the word “low-carbon.” All items already existed and the word was only changed. If there is no specific construct associated with low-carbon travel constraint, and all we do is to add the context, I don’t see much value in the developed scale. This comment goes back to my first question about how the authors identified those factors. I believe interviews with consumers should have been done to identify the low-carbon context-specific factors, instead of adopting previous factors, as this is a new study context. The authors should clear this up by explaining why the new scales differ from other and how the new scales provide value to researchers and practitioners.

Response: Thanks for your comments. As we have mentioned in this article, we first selected the item pools from literatures related to low-carbon travel context. After in-depth interview and contain analysis, and expert validation to extend and deep the concepts and final items of LCTMS and LCTCS. Please see the original section “3.1. Generating initial items” in pp. 7-9.

Other issues:

  1. I suggest providing a figure of the both second-order scales indicating those are the final models.

Response: Many thanks for your suggestion. We have added the two second-order figures as “Figure 2” and “Figure 3”. Please see pp. 19-20.

The CFI, SRMR, and RMSEA indicated adequate data to model fit for all models based on the general guidelines. To compare the first-order and second-order models, the researcher examined the NCP, AIC, and ECVI. Although the first-order models were marginally better, the second-order models were both reasonably close and significantly easier to use in research and practice. Pan, et al. [65] pointed out that when the models are all adapted, the higher-order model has the principle of simplification. Therefore, this study chooses the second-order models as the best hierarchical model for LCTMS (Figure 2) and LCTCS (Figure 3).

Figure 2. The second-order model of LCTMS.

Figure 3. The second-order model of LCTCS.

  1. I think the authors can further make the most of the data. Motivation and constraint have been often studied in a single model. The authors can further test the relationship between the two constructs as evidence of predictive validity.

Response: We agree your comment. Hence, in section “5.3 Limitations and suggestions for further research”, we have already mentioned:

 “…researchers can use the two scales to examine further the relationships between tourists' low-carbon travel motivations and low-carbon travel constraints and other variables (such as attitudes, subjective norms, constraint negotiation, and perceived behavioral control) to strengthen the theoretical basis of relevant research, such as the theories of planned behavior [5] and protection motivation [8].”

  1. Were participants aware of what ESCR stood for? Abbreviation in the item is not a good practice, even though there was an explanation, as some people might not pay attention. The authors might want to spell out.

Response: Thanks for your comment. In fact, we have done a translation and back-translation process from English to Chinese, then to English. Please see line 341 to 344.

This study adopted the translation and back-translation process [52]. We reviewed original items from English literature, translated them into Chinese, and ensured the responders understood the meanings of each item, such as ESCR.

Added Reference:

Lee, T.H.; Jan, F.-H. The low-carbon tourism experience: A multidimensional scale development. Journal of Hospitality & Tourism Research 2019, 43, 890-918, doi:10.1177/1096348019849675.

Reviewer 3 Report

Thank you for the opportunity to review this interesting article. Below are some considerations.
1. Since the topic is a niche, I propose to extend the introduction to several publications, eg Ram, Y., & Hall, C. M. (2020). The camp not taken: analysis of preferences and barriers among frequent, occasional and noncampers. Leisure Sciences, 1-24.
2. The test procedure and test description are appropriate.
3. The presentation of the results is good, but I suggest broadening the discussion.
4. Conclusions and research limitations are presented very well.

Author Response

Response Letter

Journal Title: International Journal of Environmental Research and Public Health

Manuscript Reference No. 1661699

Manuscript Title: Low-carbon travel motivation and constraint: Scales development and validation

We sincerely appreciate all the detailed comments and suggestions made by the reviewers. All comments are considered carefully and summarized as follows, and all the revisions are marked in Red in the original paper for reviewers’ readability. In addition, the revised manuscript has been sent to Wallace Language Editing Services for cosmetic edit. We believe the presentation of this revised manuscript should be more expressive and more readable than the original manuscript. Please do not hesitate to give us any comments and we also look forward to your further decision.

Responses from the authors to Reviewer 3:

Review Report (Reviewer 3)

Comments and Suggestions for Authors

Thank you for the opportunity to review this interesting article.

Response: Thank you very much. It’s our honor to present this work and get comments from you.

Below are some considerations.
1. Since the topic is a niche, I propose to extend the introduction to several publications, eg Ram, Y., & Hall, C. M. (2020). The camp not taken: analysis of preferences and barriers among frequent, occasional and noncampers. Leisure Sciences, 1-24.
Response: Many thanks for your rich suggestion. We have referred to this suggested article in introduction. Please see line 51 to 58.

On the other hand, most tourists recognize the benefits of low-carbon activities but are reluctant to plan low-carbon travel activities [10]. The key to influencing tourists' decisions is travel constraints that act as barriers and prevent them from traveling in general or traveling to the extent they would like [11]. Travel constraints can be defined as those that inhibit continued traveling, cause the inability to travel, result in the failure to maintain or increase the frequency of travel, and/or lead to negative impacts on the quality of the travel experience [12,13]. They prevent decision-makers from engaging in travel even though the motivation may exist.

Added Reference:

Ram, Y.; Hall, C.M. The camp not taken: Analysis of preferences and barriers among frequent, occasional and noncampers. Leisure Sciences 2020, 1-24. doi:10.1080/01490400.2020.1731885

  1. The test procedure and test description are appropriate.
    Response: Very thanks for your comment.

  1. The presentation of the results is good, but I suggest broadening the discussion.
    Response: Thanks for your suggestion, we have broadened the discussion. Please see line 660 to 686.

Scholars have been increasingly interested in determining the motivations and constraints behind tourists’ decisions to engage in low-carbon travel [5,8]. This study’s results reflect the motivations, constraints, and ESCR behaviors of tourists that have been proposed in other low-carbon travel literature. Kuo and Dai [5] and Horng, et al. [8], for example, have discussed environmental protection motivations, while Dickinson, et al. [44] and Dällenbach [42] discussed intrapersonal constraints, interpersonal constraints, and structural constraints.

Relevant literature has stressed that low-carbon travel contributes to sustainable tourism development [5,8]—Horng, et al. [54] drew a low-carbon travel literacy scale from the perspectives of tourism practitioners. Hsiao [68] contributed a low-carbon tourism evaluation index system for travel agencies. Lee and Jan [52] developed a low-carbon tourism experience scale to guide tourists and managers to reduce carbon emissions. However, Horng, et al. [54] only adopt potential literacy from some internal motivations. Hsiao [68] focused on establishing the industrial-level low-carbon tourism index. Lee and Jan [52] encouraged studies o focus on examing the causal structural models of low-carbon tourism experience.

Since few studies have comprehensively examined the low-carbon travel motivation and low-carbon travel constraint in the tourism industry, this study is based on a push and pull structure of tourism motives and a three-dimensional framework for constraints and then emphasizes the diversified orientation of low-carbon travel motivation and low-carbon travel constraint. This study suggested various critical factors indicators gathered from in-depth interviews and questionnaires. Furthermore, there are still many studies that continued to explore the relationships between travel motivation, constraint, and experience. Compared to past studies, this study provided the necessary concept and measurements of tourists’ low-carbon travel motivation and constraint in the low-carbon tourism context, the LCTMS and LCTCS developed in this article highlight the independent tourists' perspectives.

Added references:

Lee, T.H.; Jan, F.-H. The low-carbon tourism experience: A multidimensional scale development. Journal of Hospitality & Tourism Research 2019, 43, 890-918, doi:10.1177/1096348019849675.

Hsiao, T.-Y. Developing a dual-perspective low-carbon tourism evaluation index system for travel agencies. J. Sustainable Tour. 2016, 24, 1604-1623, doi:10.1080/09669582.2015.1136633.

  1. Conclusions and research limitations are presented very well.

Response: Very thanks for your comment.